# Risk of Tuberculosis Caused by Fluticasone Propionate versus Budesonide in Chronic Obstructive Pulmonary Disease: A Nationwide Population-Based Study

**DOI:** 10.3390/jpm12071189

**Published:** 2022-07-21

**Authors:** Iseul Yu, Sunmin Park, Se Hwa Hong, Min-Seok Chang, Seok Jeong Lee, Suk Joong Yong, Won-Yeon Lee, Sang-Ha Kim, Ji-Ho Lee

**Affiliations:** 1Department of Internal Medicine, Yonsei University Wonju College of Medicine, Wonju 26426, Korea; dbdltmf13@gmail.com (I.Y.); psmin88@gmail.com (S.P.); mim8776@yonsei.ac.kr (M.-S.C.); leeseokj@yonsei.ac.kr (S.J.L.); sjyong@yonsei.ac.kr (S.J.Y.); wonylee@yonsei.ac.kr (W.-Y.L.); sanghakim@yonsei.ac.kr (S.-H.K.); 2Department of Biostatistics, Yonsei University Wonju College of Medicine, Wonju 26426, Korea; hhdtpghk@yonsei.ac.kr

**Keywords:** inhaled corticosteroids, chronic obstructive pulmonary disease, tuberculosis, fluticasone propionate, budesonide

## Abstract

Background: In chronic obstructive pulmonary disease (COPD), inhaled corticosteroids (ICSs) are recommended for use by patients with frequent exacerbations and blood eosinophilia. However, ICSs are often inappropriately prescribed and overused. COPD studies have reported an increased risk of tuberculosis among ICS users. This study aimed to compare the risk of tuberculosis according to the different ICS components. Methods: This study was conducted using a nationwide, population-based cohort. Patients newly diagnosed with COPD between 2005 and 2018, and treated with either fluticasone propionate or budesonide, were selected. The patients were followed up until the development of tuberculosis. Results: After propensity score matching, 16,514 fluticasone propionate and 16,514 budesonide users were identified. The incidence rate of tuberculosis per 100,000 person-years was 274.73 for fluticasone propionate and 214.18 for budesonide. The hazard ratio of tuberculosis in fluticasone propionate compared with budesonide was 1.28 (95% confidence interval 1.05–1.60). The risk of tuberculosis for fluticasone propionate increased with higher ICS cumulative doses: 1.01 (0.69–1.48), 1.16 (0.74–1.81), 1.25 (0.79–1.97), and 1.82 (1.27–2.62) from the lowest to highest quartiles, respectively. Conclusion: Fluticasone propionate is associated with a higher risk of tuberculosis than budesonide. ICS components can differently affect the risk of tuberculosis in patients with COPD.

## 1. Introduction

Chronic obstructive pulmonary disease (COPD) is the most prevalent chronic airway disease worldwide, with a prevalence of 12.2% in the general population [1]. COPD is characterized by chronic and persistent dyspnea. However, patients with COPD often experience an exacerbation of respiratory symptoms, which require additional medications to alleviate symptoms. Inhaled bronchodilators, including long-acting muscarinic antagonists (LAMAs) and long-acting beta_2_-agonists (LABAs), can reduce the exacerbation rate and control the symptoms of COPD [2]. Despite using bronchodilators, some patients remain at risk of recurrent exacerbation. Inhaled corticosteroids (ICSs) are recommended for patients with frequent exacerbations and blood eosinophilia [3].

A large cohort of patients with COPD in the United Kingdom and United States reported that only 10% of patients with COPD had both ≥2 exacerbations and a blood eosinophil count of ≥300 cells/μL [4]. However, ICS-containing regimens have been inappropriately prescribed as initial maintenance therapy and are overused in COPD [5,6]. Inhaled bronchodilators are considered safe, whereas ICSs are associated with various side effects. Most studies have shown an increased risk of pneumonia in patients treated with ICSs compared with those treated without ICSs [7]. In addition, intraclass differences in the risk of pneumonia among ICS components revealed that fluticasone might cause more pneumonia than budesonide [8]. COPD cohort and population-based studies have reported an increased risk of tuberculosis because of ICS use [9]. However, it remains unknown whether there is a different risk of tuberculosis among various ICS components.

This study aimed to compare the risk of tuberculosis according to different ICS components: fluticasone propionate and budesonide.

## 2. Materials and Methods

### 2.1. Data Source 

All individuals living in South Korea are required to register with the National Health Insurance Service (NHIS). Those insured by the NHIS pay monthly insurance contributions according to their socioeconomic status and receive medical services from healthcare providers. The NHIS, as a single insurer, offers payment to healthcare providers based on the claims to the NHIS. The NHIS formed and operates the public database, the National Health Information Database, encompassing healthcare utilization data, socioeconomic and demographic information, and mortality statistics for the entire population of South Korea. The healthcare utilization database includes records of inpatient and outpatient usage (diagnosis, length of stay, treatment costs, and received services) and prescription records (drug name, days prescribed, and daily dosage). Classification of diagnosis is based on the International Classification of Diseases, 10th Revision. All records are converted to matched electronic codes in the database and offered for conducting medical research or making public health policies [10,11].

This study was approved by the Institutional Review Board of Wonju Severance Christian Hospital (CR322321) and adhered to the principles of the Declaration of Helsinki. As this was a retrospective study using anonymous claims data, the requirement for informed consent was waived.

### 2.2. Study Design

This study was conducted using a nationwide, population-based cohort. Fluticasone propionate and budesonide were selected to compare the risk of tuberculosis because they are the most widely prescribed ICS components for COPD. Therefore, all fluticasone propionate- or budesonide-containing drugs were included in this study, regardless of the commercial name or pharmaceutical company, combination with LABAs, and the device type, such as metered-dose inhalers and dry-powder inhalers. However, fluticasone furoate, another subtype of fluticasone, was excluded because its pharmacological properties differ from fluticasone propionate [12]. 

The first prescription date for fluticasone propionate or budesonide was regarded as the index date. Inhaled bronchodilators accompanied by ICSs for 1 year after the index date were classified into short-acting beta_2_-agonists (SABAs), LAMAs, LABAs, and LAMAs/LABAs. Interval periods between COPD diagnosis and the index date were collected. The patients were followed up until the development of tuberculosis. The history of comorbidities before the index date, such as bronchiectasis (J47), diabetes (E10–E14), hypertension (I10), heart failure (I11, I50), stroke (I60–I69), chronic kidney disease (N17–N19), and chronic liver disease (K70–K76), were evaluated based on the diagnostic codes. Oral corticosteroids (OCSs) might be related to the history of COPD exacerbation, thus reflecting the risk of exacerbation at baseline. The presence of an OCS prescription and duration of the OCS prescription was calculated to match the risk of exacerbation between the two groups.

### 2.3. Study Participants

COPD cases were defined using the corresponding diagnostic codes (J42–J44, except for J430) and records of pulmonary function tests before or after 1 year of COPD diagnosis. Patients who had lung cancer between 2005 and 2018 and those diagnosed with COPD before 2005 were excluded. Therefore, patients newly diagnosed with COPD without lung cancer between 2005 and 2018 were included. Among them, half of the participants were selected with random sampling and offered for research purposes according to the national policy applied to the usage of public databases. Additionally, patients younger than 40 years and those with a prescription history of both fluticasone propionate and budesonide, a history of tuberculosis before the index date, and no prescription of fluticasone propionate or budesonide were excluded. Finally, patients treated with fluticasone propionate or budesonide for at least 1 month were included in the study.

### 2.4. Study Outcomes

Tuberculosis was defined using a diagnostic code (A15–A19 and U88.0–U88.1) and the use of two or more of the following anti-tuberculosis drugs prescribed within 90 days of the first diagnosis of tuberculosis: isoniazid, rifampicin, ethambutol, pyrazinamide, rifabutin, and cycloserin [13]. The cumulative ICS dose was calculated to determine the dose–response relationship in the risk of tuberculosis. Budesonide was converted to fluticasone propionate. The equivalent ICS dose was 50 μg of fluticasone propionate and 80 μg of budesonide [13]. The cumulative ICS dose was the sum of all prescribed ICSs for the entire study period and was classified into quartiles.

### 2.5. Statistical Analysis

Propensity score matching was conducted to balance the baseline characteristics of the patients treated with fluticasone propionate or budesonide. Logistic regression analysis was used for propensity score matching and included diverse variables at baseline, including age, sex, comorbidities, bronchodilators, OCS prescription, and interval from COPD diagnosis to the index date. Descriptive statistics of the baseline characteristics of the two groups are presented using Student’s *t*-test for continuous variables and the chi-squared test for categorical variables. The hazard ratio (HR) of tuberculosis, describing the relative risk of tuberculosis with fluticasone propionate compared with budesonide, was analyzed using the Cox proportional hazards model. Significant variables in the baseline characteristics after matching were included in the multivariate analysis. All statistical analyses were conducted using SAS 9.4 (version 9.4; SAS Institute Inc., Cary, NC, USA). Statistical significance was set at *p* < 0.05.

## 3. Results

### 3.1. Baseline Characteristics

The numbers of patients treated with fluticasone propionate and budesonide were 38,628 and 16,514, respectively. Significant differences were noted in the baseline characteristics of unmatched participants (Appendix A). After propensity score matching at a 1:1 ratio, the same number of patients (*n* = 16,514) were allocated to fluticasone propionate and budesonide (Figure 1). The baseline characteristics of the two groups were compared (Table 1). The mean age was 62.39 ± 11.68 years for the fluticasone propionate group, and 62.47 ± 11.54 years for the budesonide group (*p* = 0.5205). The distribution of comorbidities, including the Charlson Comorbidity Index and OCS prescription, did not differ between the two groups. There was a difference in accompanying bronchodilator use. SABA use was lower in the fluticasone propionate group than that in the budesonide group (1.98% vs. 6.27%, *p* < 0.0001). LABA use was higher in the fluticasone propionate group than that in the budesonide group (77.91% vs. 73.48%, *p* < 0.0001). The interval from COPD diagnosis to the index date was longer in the budesonide group compared with the fluticasone propionate group (467.7 ± 905.8 days vs. 446.2 ± 830.7 days, *p* = 0.0245). 

### 3.2. Incidence of Tuberculosis

All patients were classified into quartiles, from the lowest (Q1) to the highest (Q4), according to the cumulative ICS dose (Table 2). The proportions of the fluticasone propionate group were 34.46% (Q1), 16.35% (Q2), 24.34% (Q3), and 24.85% (Q4). The budesonide group was distributed as 36.08% (Q1), 19.81% (Q2), 21.24% (Q3), and 22.88% (Q4). The crude incidence rate of tuberculosis per 100,000 person-years was 274.73 for the fluticasone propionate group and 214.18 for the budesonide group (Table 3). The incidence rates of tuberculosis were 233.04 (Q1), 296.32 (Q2), 238.36 (Q3), and 339.08 (Q4) for the fluticasone propionate group and 235.79 (Q1), 251.30 (Q2), 187.75 (Q3), and 183.51 (Q4) for the budesonide group.

### 3.3. Risk of Tuberculosis

In the unmatched participants, the HR of tuberculosis with fluticasone propionate in comparison to budesonide was 1.44 (95% confidence interval [CI] 1.22–1.70) in the univariate analysis and 1.21 (95% CI 1.02–1.44, *p* = 0.0027) in the multivariate analysis (Appendix A). In the matched participants, the HR of tuberculosis with fluticasone propionate in comparison to budesonide was 1.26 (95% CI 1.04–1.53, *p* = 0.0180) in the univariate analysis and 1.28 (95% CI 1.05–1.60, *p* = 0.0167) in the multivariate analysis (Figure 2). The HR of tuberculosis with fluticasone propionate compared with budesonide increased at higher ICS cumulative doses: Q1, 1.01 (0.69–1.48, *p* = 0.9523); Q2, 1.16 (0.74–1.80, *p* = 0.5168); Q3, 1.25 (0.79–1.97, *p* = 0.3344); Q4, 1.82 (1.27–2.62, *p* = 0.0012) (Appendix A).

## 4. Discussion

In this nationwide, population-based study, fluticasone propionate was more associated with tuberculosis than budesonide. Significant results were found in unmatched participants and were maintained after conducting a propensity score matching for possible confounding factors and multivariate analysis. Additionally, the risk of tuberculosis with fluticasone propionate compared with budesonide increased in proportion to the cumulative dose of ICSs.

The most well-known pulmonary complication of ICSs in COPD is pneumonia. All study designs, including randomized control trials (RCTs), meta-analyses, and observational studies, consistently reported an elevated incidence of pneumonia due to the use of ICSs [14]. In addition, a meta-analysis of RCTs showed an increased risk of tuberculosis, with an odds ratio (OR) of 2.29 (95% CI 1.04–5.03) in ICS users, compared with those who do not use ICSs [15]. The first observational study conducted using the health administrative database of Canada reported the increased risk of tuberculosis in any ICS users (1.27; 1.05–1.53) and current ICS users (1.33; 1.04–1.71) in patients with respiratory diseases [16]. However, they were not findings specific to COPD because both asthma and COPD were present in the included in study subjects. Another population-based study of Korea showed an increased rate of tuberculosis in ICS users compared with non-ICS users (1.20; 1.05–1.37) [13], while an observational study of the population of Taiwan did not find a different rate of tuberculosis in ICS users (1.08; 0.91–1.27) [17]. Meta-analysis of observational studies reported that ICS use was associated with an increased risk of tuberculosis, with an OR of 1.46 (1.06–2.01) [18]. Therefore, COPD guidelines recommend against using ICSs in patients with repeated pneumonia events and a history of mycobacterial infection [19].

The intraclass differences among ICS components need to be considered. Subgroup analysis of a meta-analysis comprising RCTs showed that budesonide was not associated with pneumonia risk, while fluticasone propionate was linked to increased pneumonia risk [20]. Pooled analysis of direct comparative studies reported an increased risk of pneumonia with fluticasone propionate compared with budesonide [8]. A study conducted using a health administrative database in Canada indicated that nontuberculous mycobacterial pulmonary disease was associated with fluticasone but not with budesonide [21]. To the best of our knowledge, only one study has directly compared intraclass differences in the risk of tuberculosis [22]. The study participants were extracted from a subset of Taiwan’s national health administrative database. Fluticasone propionate was associated with an increased risk of tuberculosis compared with budesonide, with an HR of 1.45 (1.21–1.74). The outcome measurement for tuberculosis was defined as the diagnostic code and medication history for tuberculosis, which was similar to that of our study. However, there was a different selection flow in the study participants than in our study. Only fluticasone/salmeterol or budesonide/formoterol in a fixed combination was included. A previous diagnosis of tuberculosis before the index date and lung malignancy were not excluded. Our study included all fluticasone propionate- or budesonide-containing drugs and excluded tuberculosis cases before the index date and lung malignancy, which might affect the development of tuberculosis. Therefore, it is assumed that different inclusion and exclusion criteria contributed to the slightly different risks of tuberculosis between the two studies.

Our study reported an increased risk of tuberculosis in proportion to the cumulative ICS dose of fluticasone propionate. A previous study conducted using the Korean national claims database showed that the risk of tuberculosis was associated with the cumulative dose of ICSs [13]. When the cumulative dose of ICSs was divided into quartiles, the ORs were 1.09 (0.92–1.28), 1.15 (0.95–1.39), 1.87 (1.45–2.42), and 2.14 (1.71–2.66), respectively, from the lowest to highest quartiles. Further comparative analysis was not performed to determine whether fluticasone propionate or budesonide affected the dose-dependent risk of tuberculosis. However, from the perspective of our study results, fluticasone propionate might have contributed to the dose-dependent risk of tuberculosis in a previous study. Additionally, our study results correspond with the strategy of ICS withdrawal if there is an inappropriate indication or lack of response to ICS [23]. Accordingly, discontinuation of ICS should be considered in patients at a high risk of tuberculosis, as this may lead to a lower risk of tuberculosis.

The increased risk of tuberculosis with fluticasone propionate could be explained by the mechanistic studies of ICSs. Fluticasone propionate has different pharmacological properties than budesonide. Budesonide is highly water-soluble, whereas fluticasone propionate is lipophilic. Therefore, the absorption rate of fluticasone propionate is very slow, whereas budesonide is rapidly dissolved in the mucosal lining fluid [24]. Fluticasone has a fluorine moiety in its chemical structure, which makes it more lipid-soluble [25]. Fluticasone propionate is more potent than budesonide in terms of immune suppression. Fluticasone propionate suppresses pro-inflammatory cytokines, such as interleukin-6 (IL-6), IL-8, and tumor necrosis factor-α, in alveolar macrophages and epithelial cells stimulated with dust or lipopolysaccharide in vitro [26]. Budesonide was ten times less potent than fluticasone propionate in suppressing pro-inflammatory cytokines. Additionally, fluticasone propionate is more potent in suppressing vascular cell adhesion molecule-1 expression in bronchial epithelial cells, leading to the inhibition of leukocyte recruitment to infection sites [27]. Accordingly, lipophilicity, prolonged absorption rate and persistence in the airway mucosa, and the potent immunosuppressive activity of fluticasone propionate substantiated in preclinical studies might affect the protective mechanism against tuberculosis.

The strength of our study was its design. This study was conducted using a nationwide claims database. It includes all populations living in South Korea. Therefore, our study participants and results effectively represent real-world clinical contexts. Additionally, we attempted to include possible covariates regarding COPD severity in propensity score, matching to control for selection bias. The present study has several limitations. First, the diagnosis of tuberculosis was based on a diagnostic code with anti-tuberculous medication history. In this observational study using claims data, we could not find results of the usual diagnostic modality for tuberculosis, such as acid-fast bacilli stain or culture and polymerase chain reaction. Second, there may have been confounding factors that interfered with the direct comparison between fluticasone propionate and budesonide. COPD- or tuberculosis-related factors, such as alcohol consumption, smoking, lung function, and immune status, were not measured because of the inherent limitations in studies based on claims databases.

## 5. Conclusions

Fluticasone propionate is associated with a higher risk of tuberculosis than budesonide. ICS components can differently affect the risk of tuberculosis in patients with COPD. Therefore, ICS components, as well as ICS itself, need to be carefully considered when initiating ICS treatment in patients at risk of tuberculosis. Further clinical and mechanistic studies of the impact of ICS components on the development of tuberculosis are warranted.

## Figures and Tables

**Figure 1 jpm-12-01189-f001:**
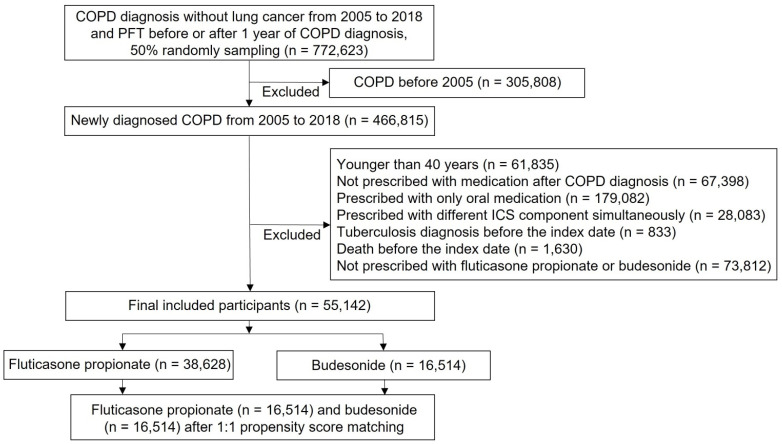
Flowchart of patient selection.

**Figure 2 jpm-12-01189-f002:**
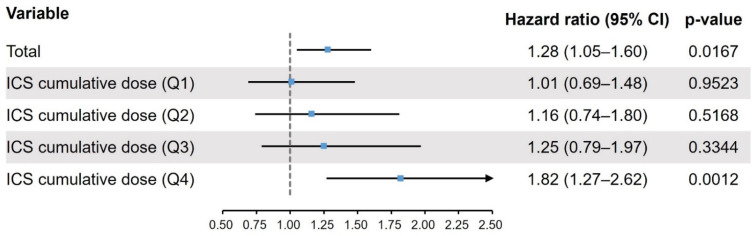
Risk of tuberculosis in fluticasone propionate users compared with budesonide users. Hazard ratios were calculated as the ratio of the risk of tuberculosis in fluticasone propionate compared with budesonide (reference) within identical doses and are presented with 95% CIs and *p*-values. CI, confidence interval; ICS, inhaled corticosteroid.

**Table 1 jpm-12-01189-t001:** Baseline characteristics of the study participants.

	Fluticasone Propionate (*n* = 16,514)		Budesonide (*n* = 16,514)		*p*-Value
	*n*	%	*n*	%	
**Age** (years)					
Mean (SD)	62.39 (11.68)		62.47 (11.54)		0.5205
40–49	2728	16.52	2590	15.68	0.2795
50–59	4188	25.36	4174	25.28	
60–69	4754	28.79	4854	29.39	
70–79	3660	22.16	3716	22.50	
≥80	1184	7.17	1180	7.15	
**Sex**					
Male	9413	57.00	9364	56.70	0.5862
Female	7101	43.00	7150	43.30	
**Comorbidity**					
Bronchiectasis	1436	4.92	1452	4.92	0.7553
Diabetes	5132	17.60	5215	17.66	0.3248
Hypertension	9346	32.05	9445	31.99	0.2713
Heart failure	3217	11.03	3260	11.04	0.5512
Stroke	3530	12.11	3569	12.09	0.6014
Chronic kidney disease	825	2.83	861	2.92	0.3681
Chronic liver disease	5674	19.46	5720	19.38	0.5944
**CCI**					
Mean (SD)	3.05 (2.11)	3.01 (2.08)	0.0615		
<2	4542	27.50	4586	27.77	0.5882
≥2	11,972	72.50	11,928	72.23	
**Bronchodilator**					
SABA	327	1.98	1035	6.27	<0.0001
LAMA	2186	13.24	2231	13.51	0.4669
LABA	12,866	77.91	12,134	73.48	<0.0001
LAMA/LABA	1135	6.87	1114	6.75	0.6464
**OCS prescription**					
Yes	13,613	84.72	13,602	84.65	0.8467
No	2456	15.28	2467	15.35	
**OCS prescription day**					
Mean (SD)	11.07 (17.37)		10.98 (17.56)		0.6521
**Interval from COPD** **diagnosis to index date**					
Mean (SD)	446.2 (830.7)		467.7 (905.8)		0.0245

SD, standard deviation; CCI, Charlson Comorbidity Index; SABA, short-acting beta_2_-agonist; LAMA, long-acting muscarinic antagonist; LABA, long-acting beta_2_-agonist; OCS, oral corticosteroid; COPD, chronic obstructive pulmonary disease.

**Table 2 jpm-12-01189-t002:** Proportion of participants according to the cumulative doses of ICSs.

	Fluticasone Propionate (*n* = 16,514)		Budesonide (*n* = 16,514)	
	*n*	%	*n*	%
**ICS cumulative dose** (μg)				
Mean (SD)	185,521 (551,181)		138,827 (314,267)	
0–15,000 (Q1)	5691	34.46	5958	36.08
15,001–60,000 (Q2)	2700	16.35	3271	19.81
60,001–225,000 (Q3)	4019	24.34	3507	21.24
>225,000 (Q4)	4104	24.85	3778	22.88

Budesonide was converted to fluticasone propionate. ICSs, inhaled corticosteroids; SD, standard deviation.

**Table 3 jpm-12-01189-t003:** Crude incidence rate of tuberculosis according to the ICS components and cumulative dose.

	Fluticasone Propionate			Budesonide		
Variables	Person -Years	Tuberculosis Cases	Incidence (per 100,000)	Person- Years	Tuberculosis Cases	Incidence (per 100,000)
Total	84,810.47	233	274.73	85,907.51	184	214.18
ICS cumulative dose (μg)						
0–15,000	27,033.70	63	233.04	28,415.55	67	235.79
15,001–60,000	13,161.30	39	296.32	15,916.96	40	251.30
60,001–225,000	20,137.46	48	238.36	17,043.64	32	187.75
>225,000	24,478.01	83	339.08	24,521.94	45	183.51

## Data Availability

Data were obtained from the National Health Insurance Sharing Service (NHISS) and are available at https://nhiss.nhis.or.kr (accessed on 23 May 2022). NHISS allows access to all of this data for any researcher who promises to follow the research ethics at some cost. Those seeking access to this articles’ data can download it from the website after promising to follow the research ethics.

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
