# Peer review of "Risk of Tuberculosis Caused by Fluticasone Propionate versus Budesonide in Chronic Obstructive Pulmonary Disease: A Nationwide Population-Based Study"

_jpm, 2022, doi:10.3390/jpm12071189_

Round 1
Reviewer 1 Report
The paper is interesting and well written.
Small correction in discussion- more explanation and discussion regarding published papers about ICS and tuberculosis.
Reviewer 2 Report
The endeavor of the authors is appreciated, as the topic seems to be relevant and promising. All the sections have been described well. However, there is some suggestion for its improvement.
Figure 1. Flowchart of patient selection: Review the calculation, newly diagnosed COPD cases from 2005 to 2018 should be 466,815
Page 156-157: The description in the text did not match table 2. Kindly mention, which values were correct.
Explore the implication and explanation of the findings. Despite the report that fluticasone propionate is associated with a higher risk of tuberculosis than budesonide. Would you like to suggest avoiding the use of Fluticasone propionate?
